# An Innovative Approach to Delivering a Family-Based Intervention to Address Parental Alcohol Misuse: Qualitative Findings from a Pilot Project

**DOI:** 10.3390/ijerph19138086

**Published:** 2022-07-01

**Authors:** Hayley Alderson, Andrea Mayrhofer, Deborah Smart, Cassey Muir, Ruth McGovern

**Affiliations:** Population Health Sciences Institute, Newcastle University, Baddiley Clarke Building, Richardson Road, Newcastle Upon Tyne NE2 4AX, UK; andrea.mayrhofer@newcastle.ac.uk (A.M.); deborah.smart@newcastle.ac.uk (D.S.); c.muir@newcastle.ac.uk (C.M.); r.mcgovern@newcastle.ac.uk (R.M.)

**Keywords:** alcohol-related harms, service delivery, family approach, early help interventions, parental alcohol misuse

## Abstract

Within child welfare systems, the issue of parental alcohol misuse (PAM) and the responsibility for supporting children affected by PAM impacts on multiple health and social care services. An innovation pilot project (IPP) was set up to reduce the fragmentation between services and to help identify children affected by PAM. The current study presents findings regarding the IPP, examining its implementation, the service delivery, and the perceived impact for family members. Qualitative data were collected from 41 participants. This included interviews with alcohol-misusing parents (*n* = 13), affected adult family members (*n* = 5), and children affected by PAM (*n* = 9). Two focus groups and three one-to-one interviews were conducted with project workers (*n* = 7) and multi-agency service managers (*n* = 7). Data were analysed thematically relating to three main themes: (1) innovation in team composition and multi-disciplinary team working, (2) innovative ways of working, and (3) the benefits of a whole-family approach. The findings highlighted the importance of time for the team to ‘bed in’ and come together under one structure, a focus and oversight on whole-family care, and the importance of offering early, targeted, and flexible interventions to prevent crisis points and manage the consequences of PAM. Consideration will need to be given to joint commissioning to strengthen family-focused support.

## 1. Introduction

Parental alcohol misuse (PAM) is significantly linked to harm to children [1,2,3,4]. Evidence shows that a child’s physical and mental health can be affected, along with the development of internalising behaviours (anxiety and depression), externalising behaviours (impulsivity and aggression), and lower educational attainment [5]. Parental drinking can result in inconsistent, impaired, or inappropriate parenting, unpredictable behaviour, and in children having to take on caring responsibilities for parents and/or younger siblings [6]. Parental alcohol misuse is often a factor in child protection cases, and when PAM is present, children often have poorer welfare outcomes [5]. Parents may delay seeking help related to alcohol misuse due to a fear of being stigmatised, fear of social service involvement, and ultimately children being removed from their care [7,8]. Within a child welfare system, the issue of PAM and the responsibility for supporting children affected by PAM impacts on multiple health and social care services.

Data submitted to the National Drug Treatment Monitoring System (NDTMS) across England showed that in 2018–2019 the estimated number of alcohol-dependent adults living with children was 120,552, and the estimated number of children living with adults with alcohol dependence was between 188,858 and 207,560 [9]. Across England between 1 April 2019 and 31 March 2020 there were a total of 131,830 new presentations to adult treatment services. Of the 131,830 new presentations, 68,269 (52%) were parents. Of the 68,269 parents, 27,873 (21%) had children living with them and 40,396 (31%) were parents whose children did not currently live with them in the same household. Of the 27,873 new presentations of parents or adults who had children living in the same household within that local authority, 20,631 (74%) had no family support. While 1571 (6%) were receiving early help, 1609 (6%) were identified as a child in need, 2749 (10%) had a child protection plan in place, and 592 (2%) were in the care of their local authority (state care). Of the 40,396 parents who do not have a child living with them in the same household, 32,735 (81%) had no family support. While 592 (1%) were receiving early help, 822 (2%) were identified as a child in need, 2102 (5%) had a child protection plan in place, and 2674 (7%) were in the care of their local authority (state care).

Taking a family-centred approach to address alcohol use is not a new concept [10], and it has been advocated in response to complex social problems, which aim to recognise the family as a unit, focus on a strengths rather than a deficits perspective, and maximise the choices available to families [11]. However, in the UK, treatment for alcohol use is traditionally commissioned through an individualistic lens, with the cause, effect, and intervention focusing upon the individual who uses/misuses alcohol [12]. Due to clear evidence of alcohol-related harm to children [13] and affected family members, there has been a growing recognition of the importance of involving family members in the treatment of alcohol users and an acknowledgment that affected family members may benefit from treatment in their own right [14]. In an attempt to address unmet needs, the Department of Health and Social Care and the Department for Work and Pensions in the UK committed GBP 4.5 million of joint funding to local authorities in 2018 (a local authority is a local government organisation responsible for the provision of public services within a geographical area) via the Innovation Fund for Children, aimed at improving the support services for children of dependent drinkers and alcohol-dependent parents. The Innovation fund was the first ever dedicated government funding in the UK to support children and families affected by parental alcohol misuse [15].

This paper reports the qualitative findings from one funded project, referred to as an innovation pilot project (IPP), which brought together health and social care services with the aim of providing a combined family approach to increase the identification of families experiencing parental alcohol misuse, provide early interventions, and improve support through reduced system fragmentation. Prior to this IPP, parents and/or children affected by PAM in the area were referred to various alcohol services, children’s services, and/or third-sector organisations directly to receive specialist support. Typically, the adults and/or children were supported separately without specific reference to PAM. It is important to acknowledge that this IPP was a multiagency project from its inception. The application submitted to the Innovation Fund was a joint application between all agencies involved in the IPP to ensure that strategic buy-in had been secured prior to the project starting. The funding allocated by the Innovation Fund for Children enabled all staff delivering interventions as part of the IPP to be seconded initially on a 2-year fixed term basis (1 October 2019 to 30 September 2021) to the IPP. In October 2021, the IPP was mainstreamed and became part of the usual care. The IPP consisted of specialist services to deliver mental health support for children via a dedicated Child and Adolescent Mental Health Service (CAMHS) worker who offered timely support to children, addressing their emotional and mental health needs; a young carers’ worker from a voluntary sector organisation (young carers support) and professionals supported parents by providing the necessary alcohol interventions via an established local adult alcohol treatment agency. The IPP also provided access to a support service for significant others in families affected by PAM, which included Community Reinforcement Approach and Family Training (CRAFT), a program that teaches individuals different approaches to addressing the problems caused by alcohol and drugs within their family. Figure 1 shows the IPP multi-agency approach.

The IPP approach was introduced, as there is evidence to suggest that taking a family-focused approach when addressing parental alcohol misuse, can improve outcomes for both the children and the individual misusing alcohol [1]. Evidence also shows that supportive relationships and support networks are key factors to recovery from alcohol dependency and that, when families are involved, treatment is more likely to be adhered to and be effective [16].

## 2. Materials and Methods

All parents and children (aged 11 years+) who received support via the IPP (*n* = 85 parents and 65 children affected by PAM) were invited to participate in this study once they had been discharged from the service. Interviews took place as soon as possible following discharge in an attempt to collect accurate reflections of how they had found the support received via the IPP. Prospective interviewees were approached by their allocated project workers and were requested to complete a ‘consent to contact’ form, which provided the researcher with a contact name and number. The researcher contacted each participant to introduce the study and, if the participant was willing to proceed, a participant information leaflet and consent form were emailed or posted, and a date and time convenient to the participants was arranged to conduct an interview. Due to the COVID-19 pandemic, all of the one-to-one interviews took place by telephone. All of the professionals involved directly in the management or delivery of the IPP were recruited into the study. Professionals were recruited from organisations as follows: local authority social care and public health practitioners (*n* = 6), alcohol treatment providers (*n* = 3) CAMHS workers (*n* = 2), CRAFT workers (*n* = 2), and a young carers organisation (*n* = 1). The multi-agency service managers and project workers who delivered the IPP interventions were contacted directly by the researcher and invited to participate in a focus group. One focus group took place face-to-face (pre-COVID-19), and one focus group took place remotely via an online platform. In addition, the one-to-one project worker interviews took place via an online platform. Participant information leaflets and consent forms were sent via email. There were separate participant information leaflets for parents, project workers, and children (one version for 11–16 years old and one for 16 years+). The participant information leaflets highlighted that concern about a person’s safety or risk of harm would lead to confidentiality being broken and that this information would be shared with the IPP professional who had worked with the family member, and the available safeguarding policies would be adhered to. This was reinforced verbally at the beginning of each interview.

Prior to each interview or focus group, written (electronic) informed assent/consent was obtained from all participants, including consent from the parent/caregiver with parental responsibility for participants under 16 years of age. This was returned via email.

### 2.1. Data Collection

During recruitment, we used purposive sampling. Our only inclusion criteria were that parents, affected family members. and children had to have directly received support from the IPP and be able to provide informed consent. Professionals had to be involved in the strategic management or the delivery of support via the IPP. Data were collected from a total of 41 participants. This included 13 interviews with parents and 5 with affected adult family members (husband/wife of the adult drinker). The adult participants ranged from 19 to 55 years of age; two were male, and sixteen were female. For parents and affected family members, the average length of time for support was 6 months. Nine interviews were conducted with children affected by PAM. The children ranged from 11 to 21 years; one was male, and eight were female. For children accessing support via CAMHS, the average length of time for support was 9 months, and for children accessing support via young carers, the average length of time for support was 3 months. Interviews with participants who had accessed the IPP interventions explored their experiences of service delivery, the content of the sessions, and the outcomes.

Two focus groups and three one-to-one interviews were conducted with professionals. The first focus group with professionals (*n* = 6) and a one-to-one interview with one practitioner took place in January 2020 during the initial setup of the IPP. This data collection phase explored the challenges and facilitators of the initial setup and the hopes for the IPP as it became embedded. A second focus group with professionals (*n* = 5) took place in April 2021 to explore their views on how the project had impacted on service user care, the experiences of the referral pathways, and plans regarding the sustainability of the project. Finally, two one-to-one interviews were conducted in October 2021 with professionals working in the Multi-Agency Safeguarding Hub (MASH) team to discuss their understanding of the IPP service offer and their experiences of using the referral pathways.

All interviews and focus group discussions were semi-structured, which enabled the direction of the interview to be guided by the pre-set topic guide while also allowing the interviewer to explore emergent issues. Data collection was carried out by experienced researchers (authors HA, DS, and CM), audio-recorded, and then transcribed verbatim by a professional transcription service. The transcripts were checked by the researcher who conducted the interview for accuracy. Transcripts were anonymised, and a participant key was stored separately to maintain anonymity. Parents and children affected by PAM were given a GBP 10 voucher following the completion of an interview to demonstrate that their contributions were valued and their expertise respected [17]. Ethical approval was obtained from a North East Research Ethics Committee in November 2019 (19/NE/0294).

### 2.2. Data Analysis

The qualitative interview transcripts were analysed thematically to identify themes within the data [18]. The initial coding organised data corresponding to the research questions. Therefore, data were coded in relation to referrals, which organisations were involved in delivering the intervention, the format and content of sessions, and how the IPP differed from other services [19]. Subsequent coding focused on emerging themes [20,21], which were discussed and refined [22] with the research team. As findings emerged, they were shared within the bimonthly academic study team meeting to aid the development of ideas. The findings were also shared within the IPP steering group inclusive of practitioners from across social care, health, and criminal justice domains to provide an opportunity for professionals to identify areas of importance to them and to consider areas of further exploration. Datasets from parents, affected family members, children affected by PAM, project workers, and service managers were imported into QSR NVivo12 [23] to facilitate data management during the coding and data analysis processes. Anonymous respondent quotes are reported with ID numbers.

## 3. Results

The three main themes that arose from the data were: innovation in team composition and multi-disciplinary team working, innovative ways of working, and the benefits of a whole-family approach.

### 3.1. Innovation in Team Composition and Multi-Disciplinary Team Working

The IPP introduced and embedded a multiagency response to increase the identification of families affected by parental alcohol misuse by initiating clear referral pathways and the provision of evidence-based interventions and support. Several enablers and challenges associated with this innovative way of working were identified, such as establishing a multi-agency team, using a combined Liquidlogic Children’s Social Care System (LCS) case management recording system, being based with the MASH team, and the potential for joint commissioning. They are described in further detail below.

An initial challenge that IPP team members discussed was that while the goal of the project was to create a team that brought together individuals with varying levels of expertise and complementary skills, the way the project achieved this was by seconding staff from different organisations. However, this introduced the need for a ‘bedding in’ period and highlighted that the IPP needed to clarify role descriptions, responsibilities, and lines of accountability. One service manager commented on the importance of joint meetings with service managers and project workers, especially at the beginning, to conduct clear mapping exercises in relation to the adapted ways of working:
“*…staff seconded from a local adult alcohol treatment agency were used to delivering group sessions and had huge caseloads, so going into families’ homes and working one-to-one was very different…having to find out what their support needs are. Parents and children tend to present very differently in a facilitated group session or at a children’s centre compared to when meeting them in a family environment. At the IPP, instead of a paper-heavy assessment involving really personal questions, we started building relationships. We changed how we worked. We focus on building a relationship first and then ask these very difficult personal questions later. We have more success and more honesty*”.(project worker, ID7)

Overall, IPP workers reported that they would have benefitted from a longer period of time to enable further collaborative work to be completed with professionals at all levels of the pathway regarding the development of IPP structures and to provide additional support while project workers transitioned into and became familiar with the new ways of working.
“*I think we had quite a short lead in, really, into getting the project up and running, so learning for me, moving forward…… If you want people to work very, very differently you’ve either got to let hiccup time happen or have better, maybe, lead in time and prep*”.(project worker, ID4)

From the IPP workers’ perspective, there was also a need to clarify the structures of accountability. When the project started, project workers who delivered the interventions as part of the IPP were seconded from a range of agencies and services, as depicted in Figure 1. Initially, supervision was conducted by their original employing organisations. However, that was problematic due to the project workers day-to-day work being overseen by a manager within the IPP team. As the IPP became embedded, staff supervision was taken over by a project coordinator, which was a newly established role. The project coordinator took responsibility for the allocation and oversight of the IPP’s caseload as a part of operational management. The introduction of this role was conducive to promoting a clearer team identity, providing coherence about the IPP aims, and contributing to stability within the team after the initially high turnover of staff.

However, once embedded, the team composition was perceived as an integral element of the project’s success. Professional backgrounds included safeguarding, working with children, providing support regarding mental health and support to young carers, working with adults misusing substances, delivering training programmes around alcohol misuse (CRAFT training), and staff completing early help assessments. The project co-ordinator was situated within a MASH. This provided an opportunity for families to be identified by a wide range of concerned organisations.
“*The majority of our contacts come in from the police, but we also get a lot of contacts from schools, from health. We get anonymous referrals. Anybody and everybody can refer into front door*”.(project worker, ID14)

In addition, one of the early decisions made was to have engagement with any element of the IPP recorded on the LCS case management system so all IPP workers could see when a family was engaging (or not) with the IPP. Having all the information together on one system enabled the IPP team to clearly see which support had been offered, whether it had been accepted/declined, and the levels of contact and attendance. This allowed for a more holistic approach to be taken to the service offer. As one of the managers explained:
“*…when a Police-CCN (child concern notification) comes through, everybody can see what is happening and who is involved. Things don’t get missed and conversations can happen. We can have a co-ordinated response. This would not have been possible before*” .(project worker, ID4)

This is particularly important because communication via a range of different services can be fragmented. Being seen by different organisations or members of staff can create misunderstandings and add to tensions rather than resolve them. Recording community engagement on the local authority case recording system helped to establish a continuity of narrative and interaction, which is critical when considering interventions and/or treatment options.
“*The communication—so being able to use things like Teams and the LCS to communicate all of the issues rather than external emails and all of that sort of thing. It just makes it a lot smoother, a lot easier*”.(project worker, ID14)

The ability for the IPP to consider and/or address multiple presenting issues for an individual and the wider family prompted discussions around the joint commissioning of services. Project workers felt that alcohol misuse is usually not a stand-alone issue and other services need to be jointly commissioned, as expressed by one project worker:
“*there’s always something else going on, whether it [alcohol misuse] has been a coping mechanism and there’s now financial issues… when we’re working with people, we can decrease their alcohol level, but if they’re not getting the support for their mental health, we’re never going to really achieve anything significant…this is where co-commissioning element comes in……the mental health services are critical*”.(project worker, ID7)

Alongside the unique team composition described above, participants described the adapted and inventive ways of offering support to families affected by PAM.

### 3.2. Innovative Ways of Working

One of the perceived benefits of the IPP was that it provided an opportunity to offer early intervention to families and introduced the potential to conduct preventative work with families:
“*I think for schools especially, they know stuff, but they don’t know where to go with it. If it’s not safeguarding, yes, they can do an early help assessment but if the parent doesn’t want to talk about it or doesn’t acknowledge what the concerns are, they’re almost stuck. Where now they’ve got a community resource, if you like, that they can refer into and they can support a family with. They’ve got a different way of approaching it at an early stage*”.(project worker, ID11)

The opportunity to offer interventions to family members presenting with alcohol and/or mental health needs at a level that would not usually meet the eligibility criteria for support via statutory drug and alcohol services or CAMHS was recognised and appreciated.
“*The great thing about [the IPP] is that they’ll work with a family before we get to that point, so the interventions there. That’s great, that’s ideal. You’re putting out small fires when you’re referring them into [the IPP] to do that low level intervention before, like if they’re at crisis point*”.(project worker, ID13)

The IPP had the ability to work with individuals whose alcohol use may not be at dependency levels, however, is still problematic in terms of parenting ability, inappropriate behaviour when binge drinking, and/or alcohol use increasing levels of parental conflict. The IPP was valued, as many parents would not recognise that they needed to access specialist support. The IPP was able to promote the support as an intervention that might improve the functioning of the family unit as part of an ‘early help’ package.
“*The main service is drug and alcohol, I’ve heard a lot from my clients that when they’ve been to [statutory service], they feel like a fish out of water because there’s so many different kinds of substances used there, and people that perhaps are more dependent or using substances that they deem as kind of worse than what they are, so they then feel that they don’t need the service whatsoever, and then they kind of fall by the wayside and end up drinking more*”.(project worker, ID12)

The flexibility of the IPP granted project workers with more autonomy in terms of arranging appointments. Parents and children affected by PAM were given the opportunity to choose the locations of their appointments, including being seen within the home or at a community venue. This contrasts with usual care when individuals are expected to attend a specialist service or clinical setting. Having the flexibility to offer support within the family home enabled participants to operate around their work commitments, children’s schooling, running errands, and keeping other appointments. Appointments taking place within the home (for both children and parents) enabled workers to observe the environment families were residing in. This could present opportunities to open up conversations and/or challenge individuals if information appears to be incongruent with the environment.
“*We’re seeing the whole families together, which I think is the right way. Taking a child out of that setting, seeing them in isolation in a clinic, you’ll not get the full picture in any shape or form. Going into the family home, going into schools, going to the YMCA with them, by going where they are, you’re getting to see their life and all the bits of the jigsaw start coming together*”.(project worker, ID6)

Numerous participants stated that they would have found going to a ‘specialist alcohol service’ for meetings daunting due to its association with other problematic substance use. This was particularly evident if they were not drinking themselves but received support as a partner of someone with alcohol dependency.
“*I don’t drink, I don’t do drugs, I don’t do nothing like that so I think honestly I would have been intimidated to go somewhere like that” (a specialist substance misuse service)*.(ID1, mother, ex-partner misused alcohol)

From the professional’s perspective, they acknowledged the stigma that some individuals associated with attending a specialist service. They also acknowledged that addiction could happen to anyone, and it is important for services to be accessible and family-friendly.
“*I think that’s such a stigma around addiction and we always assume that if you’re addicted to something, then there must be something really wrong with you…. But you know, addiction, it can happen straight away to anybody*”.(project worker, ID13)

However, the family home did not work so well for all family members due to distractions, for example “*children giggling all the time*” (ID6, mother who received support due to her husband’s drinking). This example reinforces that the flexibility in approach is pivotal and needs to be assessed on an individual basis.

Flexibility was also extended to children, some of whom were seen at school, either instead of a lesson or in between lessons, or at home if preferred. One young person felt that having to deal with PAM in addition to school pressures was getting to be too much. Teenagers and young people found support via the IPP to be “…*a good outlet to just to be able to talk to someone*” (YP1, girl, age 19), and provided an opportunity to “…*go over what was happening at home*” (YP2, girl, age 15).

The school setting provided a ‘safe’ environment for young people and IPP workers to meet. However, while children and young people appreciated working with IPP workers, they did not all like to be seen in a school environment and having to miss lessons, as the other children would ‘keep asking questions’. In such cases alternative arrangements could be made. The flexibility of the service enabled young people to be seen in a location they felt most comfortable with, therefore increasing the chances of them engaging in support.
“*we’ve got that freedom of being able to meet them outside of school, so doing sessions at the park or walking along the seafront, so that it’s not so much of an organised thing; they don’t have to come into a clinic, which again a lot of young people would struggle with that environment and itself would be quite difficult*”.(project worker, ID10)

In addition, the team had the capacity to offer support in a way that could accommodate a child’s needs; appointments could be offered in an accessible way not dictated by restrictions such as ‘three strikes and you’re out’. IPP workers tried to promote an open-door approach to accessing support.
“*Yes, and I think because we can approach it in more of an informal way, like, “This is your choice”, we give them the choice, we don’t force them and we don’t tell them that they have to do it, it’s just an option for them. So, it might be that initially they say no, but then two months later, three months later, they come back, and they’re interested again*”.(project worker, ID11)

The increasingly flexible ways of working that were introduced have highlighted a positive change when engaging families:
“*One of the things that seemed to have come through strongly within [the IPP] was the benefit of home visits and being in the community and going to those vulnerable adults*”.(project worker, ID5)

### 3.3. Benefits of a Whole-Family Approach

Multiple parents explained that the support offered was very context-specific and took all members of the family into account. This was perceived as different from the support they had received previously. Services were offered concurrently but individually.
“*Within a family if you’ve got one person that’s got a substance misuse, alcohol misuse, it affects the whole family. So, you go in and you fix one, you’ve still got a lot of other people within that family who are hurt, there might be trauma, so it needs to be that when you’re going in, you’re working with the whole core family*”.(project worker, ID13)

Integral to the IPP approach was offering support and having the opportunity to work with more than one member of a family simultaneously. In two cases, adult interviewees explicitly described becoming known to the project because the IPP had supported a child based on a referral via a school. Due to the project’s multi-agency approach, project workers were able to arrange support for individual family members based on each person’s specific needs and their willingness to engage with the team. For example, in one family, the interviewee was supported by a worker from the IPP and also supported by a partner agency to access CRAFT, and the children were seen by the children’s worker (CAMHS) while the father completed a rehabilitation programme through a third-sector organisation. As stated by the interviewee, they received:
“*…Support around the family… [there was] a team around the family*”.(ID6, received support for mother, children, and alcohol-misusing father)

In a separate interview a participant commented that:
“*….so, they [IPP staff] treat us like a family*”.(ID1; mother and children supported due to ex-partner’s alcohol misuse)

Due to the IPP employing a worker to provide young carers support and a specialist young people’s mental health worker, group sessions could be offered, and peer-to-peer support was available to young people, albeit intermittently due to COVID.
“*I’ve got two young people who I’ve referred in the service, I’ve seen the difference that it makes for them first-hand. These are the young people going along to the groups, having something separate from home, where they go, and they know there’s an understanding of why they’re there. There are other young people who are in a similar situation. They get to understand why there’s an addiction and why that might happen, but also just to have that little bit of fun as well and they feel normal as well around peers their own age*”.(project worker, ID13)

Coordinated support via the IPP meant that project workers from participating agencies were aware of the family’s circumstances, which obviated the need to explain their background repeatedly. However, while multi-agency support was available, individuals could elect not to accept additional support and could choose to work with the IPP only if that was their preference.
“*We had a family where Mary and Sophie were working with the same family, and they’ve done the early help assessment, but there were kind of issues around relationships within the family, so then they were referred into a mediation service so that they could do that piece of work while Sophie and Mary still could focus on the individual work they were doing with the parent and with the young person*”.(project worker, ID8)

Having an allocated worker to create a ‘safe space’ to discuss their parents alcohol use had the potential to provide support to young people, and in the case of YP5, it helped to re-establish trust between him and his formerly alcohol-misusing father. As stated by the child’s mother:
“*I think the responsiveness of it [the IPP project] was really part of why it was successful, and I think that being able to offer a space for children to be able to share their emotional distress after a particular crisis point is... If that’s offered at the right time in the right way, it’s so important to family recovery*”.(ID16, ex-partner formerly misused alcohol)

This was echoed by professionals who recognised that being able to support the whole family unit could result in positive outcomes:
“*there’s one particular family me and Mary have worked with, where we’ve worked quite closely and we’ve had meetings, all four of us, haven’t we, where they’ve been able to build those relationships back up again that had been damaged by the drinking*”.(project worker, ID12)

The IPP also worked with partners and/or ex-partners and connected family members to specialist services via multi-agency working. A range of configurations for support were established, depending on each family member’s need. Bespoke approaches to helping individuals within the family unit included helping them to analyse their situation, working through the root causes of alcohol misuse, and flagging support options. Due to the IPP supporting and/or engaging multiple members of the family, there was the potential for different perspectives to be sought regarding how parental alcohol misuse was affecting the family unit.
“*I’ve had a lot of situations where I’m being told one thing and actually through speaking to other professionals, or speaking to other family members, even the young person or the partner who’s not drinking, we would find out that actually it’s not how it seems and that can be tackled much sooner, rather than if I’d just had that drinking parent’s side of things*”.(project worker, ID12)

From the children’s perspective, the importance of being offered and receiving an intervention in a timely manner to meet *their* needs, independent of the needs of their parents, was emphasised. Despite appreciating the support once it was in place, at times children expressed that it had come too late.
“*…for me it was a bit late in the journey as the sessions took place after my mum had begun to get help and things were improving*”.(YP1, Female; age 19)

Similarly, another young person recognised that her need to access support had subsided due to their parent’s alcohol use being addressed. It was stated that:
“*…my mom is hardly drinking now, and I feel a lot better in myself….so there is no need to speak to anyone*”.(YP3, female; age 17)

Two children suggested that the IPP should be advertised in schools so that young people become more aware of the support that is available (YP6, female, age 21, and YP7, female, age 15).

## 4. Discussion

The main themes highlighted by both the interviewees who had received the intervention and by the IPP team were the project’s focus on the family and having the ability to take a whole-family perspective, being able to offer early-help interventions to prevent crisis points, and having the capacity to offer bespoke and flexible approaches to service delivery. As identified above, the IPP was implemented by staff trained in adult, child, and specialist alcohol work. There was an appreciation of the adverse impact that parental alcohol misuse can have on children’s mental health and well-being, and therefore a team was established to meet the needs of all members of the family unit. The decision to place the IPP co-ordinator within a MASH was novel. There was an awareness that social care professionals based within a MASH or early help team could be crucial to help identify families requiring support regarding PAM.

While the innovative approach aimed to improve collaborative working so that concerns could be identified earlier and managed more efficiently, as identified by professional participants, the introduction of project workers with different models of professional identity was not without challenges. In keeping with the previous literature, when substantial changes take place regarding interdisciplinary working, staff require support throughout the process [24]. Due to restricted timeframes for implementation, the IPP experienced a short lead-in time, and if we consider Tuckman’s four-stage model of small group development, ‘Forming, storming, norming and performing’ [25], the IPP staff identified with all of these stages. Professional participants recognised that challenges arose in the ‘forming’ phase when individuals were still trying to establish how they would work together. Participants reflected on the desire for a longer lead-in time for the project. The ‘storming’ phase witnessed team members experiencing uncertainty regarding boundaries and expectations in terms of new ways of working, for instance, community-based rather than clinical, family-focused rather than one-to-one work, and changes in the provision of clinical supervision and line management. The introduction of a team co-ordinator brought clarity to the IPP, and an appreciation of each project worker’s strengths and assets was reached. The ‘norming’ stage was reached when project workers recognised the potential of the new way of working and had a stronger commitment to the project’s goals.

Due to the IPP’s family focus, project workers had the capacity to offer holistic support that recognised and supported the parent misusing alcohol, their non-alcohol-misusing partner, and their children. It has been identified that family members can be instrumental when motivating individuals misusing alcohol to enter treatment [26], facilitating the maintenance of abstinence [27], and when family members are in receipt of support they are taught mechanisms to cope more effectively with their own problems, which in turn will improve the environment in which the alcohol-misusing parent resides [28]. Participants within this study described how the IPP provided an environment where children could not only gain an understanding of addiction but could also be supported to undertake mediation work and be supported to rebuild relationships that had been damaged by a parent’s drinking. It was perceived that the IPP’s timely interventions and support helped to prevent crisis points and longer-term alcohol-related harms experienced by families.

The IPP offered support on a voluntary basis, and previous literature acknowledges that when help seeking and engagement with support takes place on a voluntary basis, there is less resistance, and it is more conducive to facilitating co-operative working relationships [29]. In addition, the flexible approach taken by the IPP helped to circumvent potential barriers of having to attend a specialist service for support regarding their alcohol use. It is well-documented that perceived or experienced stigma is an identified barrier to overcome when accessing substance misuse support [30], especially for mothers who experience additional concerns regarding the fear of losing parental rights due to their drinking [31]. Literature has identified that other barriers can include individuals not knowing about or being able to access treatment and the financial or time burdens of treatment [32,33]. The IPPs position within the MASH provides an opportunity to introduce the service to families who may be unaware of its existence, and the ability for family members to be seen in a variety of community settings helped to minimise potential obstacles to treatment, such as concerns about anonymity and difficulties regarding travel. In addition, the IPP was not a time-limited service. This contrasts with many specialist alcohol services within the UK who offer time-limited care, typically up to 12 weeks of duration [34]. Therefore, there is the potential for the IPP interventions to go some way to addressing some of the help-seeking barriers.

When accessing support, a therapeutic alliance was deemed to be important. It has been recognised that a therapeutic alliance can act as a mediator to change [35]. Taking a relational-based practice approach places respect and an absence of judgement at the core of any work undertaken and posits that effective relationships are central to successful outcomes [36]. Additionally, many authors report that professionals providing support in a non-stigmatising and non-judgemental way encourages help seeking and promotes ongoing engagement [37,38]. This finding was reinforced by parents and their children who positively reported on IPP workers being good listeners, non-judgemental, honest, helpful, and professional, all of which helped to establish a positive working relationship.

### Limitations

The main limitation experienced by the IPP was the fact that a few months after the initial implementation COVID-19 restrictions came into place. This affected service delivery in terms of having to restructure how interventions were delivered and may also have affected the access to interviewees as part of the study.

## 5. Conclusions

The IPP was designed to take a holistic, family-based approach to delivering bespoke interventions and supporting parents and children affected by PAM who often did not meet the eligibility criteria to access mainstream services. The IPP’s ability to offer early-help alcohol and mental health support to multiple family members was welcomed and was perceived to have the potential to prevent crisis points and longer-term harms experienced as a result of PAM. At the systems level, the genuine multi-agency approach was supported by professionals being seconded into the IPP project team, working from a shared location, and using a shared case management system to record interactions, which enabled communication between practitioners to take place freely. The findings of this study are worth consideration when contemplating alcohol service delivery approaches across local authorities in the UK.

## Figures and Tables

**Figure 1 ijerph-19-08086-f001:**
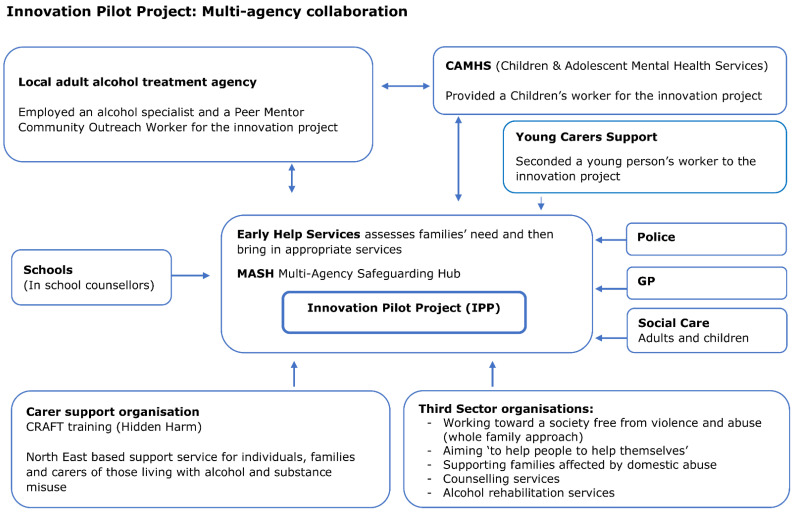
Innovation pilot project multi-agency collaboration.

## Data Availability

The datasets used and/or analysed during the current study are available from the corresponding author on reasonable request.

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
