# Peer review of "An Innovative Approach to Delivering a Family-Based Intervention to Address Parental Alcohol Misuse: Qualitative Findings from a Pilot Project"

_ijerph, 2022, doi:10.3390/ijerph19138086_

Round 1

Reviewer 1 Report

The article concerns activities carried out in the spirit of family cantered approach /family centered practice / family centered medicine (in the literature, several different wordings of this model can be found). This is the trend that has been developing since the end of the 1980s with quite extensive literature. However, the authors do not refer to this approach at all and this is probably why they present the described activities as something completely innovative. It is necessary to supplement the introduction with a description of the previous experiences of other countries / communities with the application of family centered model of work.

The research methodology is not described clearly enough. There is no information about:

- how many professionals took part in the project (preferably broken down by the institutions from which they were recruited)

- what was the criterion for selecting the respondents (both from professionals, parents and children)

- What were the initial assumptions regarding the size and characteristics of the studied sample

- Has the article presented all the content that appeared in the respondents' statements.

There are also significant gaps in the description of the IPP itself, for example:

- how long were the activities implemented (how much time elapsed from IPP initiation to collecting information on them)

- how representatives of various institutions were prepared to work in IPP (trainings?)

- what area IPP covered and how were managers of all local institutions convinced to delegate some employees to IPP

- how professionals participating in IPP reconciled their routine duties with new tasks and how their colleagues not involved in IPP approached the fact that, for example, they spend less time in the office (on their routine work).

Ambiguities regarding the program and the research methodology affect the credibility of the presented results and conclusions.

Author Response

Reviewers Comments

Authors Response

The article concerns activities carried out in the spirit of family cantered approach /family centered practice / family centered medicine (in the literature, several different wordings of this model can be found). This is the trend that has been developing since the end of the 1980s with quite extensive literature. However, the authors do not refer to this approach at all and this is probably why they present the described activities as something completely innovative. It is necessary to supplement the introduction with a description of the previous experiences of other countries / communities with the application of family centered model of work.

Thank you for your comment, I have added further information to lines 58-67 and 72-74 to highlight that this is not currently usual practice within the UK.

The research methodology is not described clearly enough. There is no information about:

- how many professionals took part in the project (preferably broken down by the institutions from which they were recruited)

I have added further information regarding organisations that professionals worked in/were recruited from on lines 118-122.

What was the criterion for selecting the respondents (both from professionals, parents and children)

I have added further information regarding how participants were selected on lines 108-112 and 118-122

What were the initial assumptions regarding the size and characteristics of the studied sample

As this was a pilot study, we did not have assumption regarding the sample size and characteristics of the studied sample in terms of the parent and children- we recruited purposefully as clarified on lines 138-141

Further demographic information added and information regarding length of support provided on lines 142-151

Has the article presented all the content that appeared in the respondents' statements.

No, it has presented the main qualitative findings, I have added Qualitative into the title and on line 172 to reinforce/clarify this.

There are also significant gaps in the description of the IPP itself, for example:

- how long were the activities implemented (how much time elapsed from IPP initiation to collecting information on them)

I have added further information to clarify the duration of the IPP, line 86- 89.

Also further information added regarding parent and children participants- they were eligible as soon as they has completed the agreed work with the IPP, lines 94-96

Further information added regarding the purpose of the interviews at different stages on lines 154-161.

How representatives of various institutions were prepared to work in IPP (trainings?)

I have added further information to clarify that the IPP was a joint application, strategic buy in was obtained prior to the project commencing, clarification provided lines 82-89

what area IPP covered and how were managers of all local institutions convinced to delegate some employees to IPP

It is not possible to identify the area that was covered by the IPP due to confidentiality.

It was a vested interest for multi-agency partners to be involved in the IPP and all agencies were jointly involved in the application for funding as clarified on lines 82-89

how professionals participating in IPP reconciled their routine duties with new tasks and how their colleagues not involved in IPP approached the fact that, for example, they spend less time in the office (on their routine work).

Ambiguities regarding the program and the research methodology affect the credibility of the presented results and conclusions.

Thank you for raising this concern. However, it is not applicable to the IPP, workers were seconded into the IPP, this has been clarified on line 82-89.

Reviewer 2 Report

Overview:  Overall, this is a very thorough report on the findings of an intervention protocol focusing on parental alcohol misuse and the qualitative reports of the initial preliminary study.

Abstract:  Provides a good overview of the background and findings of the study and a brief statement of the future of the project.  Overall the abstract is well constructed and thorough.

Introduction: 

Overall, very clear background regarding the problem and the existing services for alcohol misuse and the reasons behind the funded project.  The authors also provide good description of the specific services that were evaluated for this paper.

Line 31:  Please clarify the term “behaviour and educational attainment”

Materials and Methods:

Overall very thorough as to the process of interviews and data analysis options. 

Line 117: The letters HA, DS, and CM are used in parentheses presumably to indicate which authors were involved in the data collection.  It would be helpful to have their last names and first initials instead.  If the abbreviated versions are to be used later, then provide the letter abbreviations after the names are stated.

Results: 

Results were clearly described with relevant quotes from interviews to support their findings.  All findings were addressed in full with adequate examples for each major theme. 

Discussion, Limitations, and Conclusions:

The authors do a good job of presenting the findings in addition to contextualizing the data and results within the pandemic restrictions.  Very well written and thorough. 

Author Response

Reviewer comments

Authors response

Overview:  Overall, this is a very thorough report on the findings of an intervention protocol focusing on parental alcohol misuse and the qualitative reports of the initial preliminary study.

Abstract:  Provides a good overview of the background and findings of the study and a brief statement of the future of the project.  Overall the abstract is well constructed and thorough.

Introduction: 

Overall, very clear background regarding the problem and the existing services for alcohol misuse and the reasons behind the funded project.  The authors also provide good description of the specific services that were evaluated for this paper.

Line 31:  Please clarify the term “behaviour and educational attainment”

I have added additional information as requested to lines 31-33

Materials and Methods:

Overall very thorough as to the process of interviews and data analysis options. 

Line 117: The letters HA, DS, and CM are used in parentheses presumably to indicate which authors were involved in the data collection.  It would be helpful to have their last names and first initials instead.  If the abbreviated versions are to be used later, then provide the letter abbreviations after the names are stated.

Thank you for your comment. It is not possible to add in names currently due to blind review. It has been clarified that the initials correlate with the authors of the paper, this will become obvious once the paper has been published- please see line 165

Results: 

Results were clearly described with relevant quotes from interviews to support their findings.  All findings were addressed in full with adequate examples for each major theme. 

Discussion, Limitations, and Conclusions:

The authors do a good job of presenting the findings in addition to contextualizing the data and results within the pandemic restrictions.  Very well written and thorough. 

Thank you for your comments.

Reviewer 3 Report

I enjoyed reading this paper on an important topic.   For the most part it is well designed and reported but I would like to know if certain safeguards were introduced to enhance the robustness of the qualitative methods.  If not then this should be acknowledged as a limitation in the study.

1.  Was there an interview schedule (s) to ensure that participants all received the same or similar questions (thematic discussions).  This should be included as part of the methods section or included as an appendices.

2.  Was there any member checking of the interview transcripts?  What steps were taken (if any) to ensure accuracy of the interview transcripts.

3.  Were steps such as reflective diaries or discussions with the research team to minimise researcher influence/bias on the data?

4. Were any other safeguards put in place (probably as part of the ethical approval process to safeguard the children of the substance users.

Author Response

Reviewer comments

Authors response

I enjoyed reading this paper on an important topic.   For the most part it is well designed and reported but I would like to know if certain safeguards were introduced to enhance the robustness of the qualitative methods.  If not then this should be acknowledged as a limitation in the study.

1.  Was there an interview schedule (s) to ensure that participants all received the same or similar questions (thematic discussions).  This should be included as part of the methods section or included as an appendices.

Yes a semi structured topic guide was used- I have added in information to clarify on lines 162-164.

2.  Was there any member checking of the interview transcripts?  What steps were taken (if any) to ensure accuracy of the interview transcripts.

Thank you for this comment. The transcripts were checked by the researcher who conducted the interview for accuracy, sentence added line 166-167.

3.  Were steps such as reflective diaries or discussions with the research team to minimise researcher influence/bias on the data?

Yes, discussion took place regarding research data. I have added further information for clarification line 177-182

4. Were any other safeguards put in place (probably as part of the ethical approval process to safeguard the children of the substance users.

Thank you for bringing this to my attention. I have added information for clarification line 127-133

Round 2

Reviewer 1 Report

Thank you for the corrections / additions made. I no longer have any critical comments :-)